# Parents’ Beliefs about Children’s Emotions and Children’s Social Skills: The Mediating Role of Parents’ Emotion Regulation

**DOI:** 10.3390/children10091473

**Published:** 2023-08-29

**Authors:** Maria Cenușă, Maria Nicoleta Turliuc

**Affiliations:** Faculty of Psychology and Educational Sciences, Alexandru Ioan Cuza University of Iasi, 700554 Iasi, Romania; maria.cenusa@student.uaic.ro

**Keywords:** parents’ beliefs about children’s emotions, children’s social skills, parents’ emotion regulation, common fate model with mediation

## Abstract

Few studies have investigated the relationship between parents’ beliefs about children’s emotions and children’s social skills. Fewer studies have addressed this association and its underlying mechanisms while obtaining data from both parents. In this context, the present study explores the mediating role of parents’ emotion regulation in the association between parents’ beliefs about children’s emotions and children’s social skills. The participants were 90 parental dyads (N = 180) with typically developing preschool children. They completed self-report scales regarding parents’ beliefs about their children’s emotions, parents’ emotion regulation, and children’s social skills. The data were analyzed using the common fate mediation model (CFM with mediation). The results indicate that only parents’ cognitive reappraisal mediates the relationship between their emotion-related beliefs and their children’s social skills. Specifically, parental beliefs about “children’s anger is valuable” and “children use their emotions to manipulate others” are directly and negatively associated with children’s social skills, and indirectly through the parents’ cognitive reappraisal. Future intervention programs should focus on restructuring parents’ beliefs and their ability to regulate emotions.

## 1. Introduction

Social skills are observable indicators of social competence construct [1]. Social competence is defined as “effectiveness in interaction” and comprises intrapersonal (e.g., self-control) and interpersonal skills (e.g., cooperation) that provide both achievement of personal goals and maintaining positive interactions with others [2,3,4]. During preschool years, social skills help children manage their emotions in different contexts, initiating and maintaining relationships with peers and adults [5]. Therefore, social skills predict children’s positive outcomes in day-to-day preschoolers’ lives, playing a crucial role in their welfare and predicting a series of future positive socio-emotional outcomes [6]. For example, a longitudinal study has shown that preschoolers’ prosocial behavior, a component of social competence, was associated with young adults’ future wellness (e.g., the absence of externalizing and internalizing problems, and success in education and employment) [7].

The current study focuses on preschool children because this age category involves kindergarten attendance, a formal setting in which children are more aware of their social skills through peer interactions. At the same time, children aged 3 to 6 years spend an amount of time with parents, and through modeling, reinforcement, and imitation, parents influence children’s social skills [8,9,10]. According to the Bioecological Model [11], proximal processes (e.g., interactions with family, school, and peers) are the main engines of people’s social development. Further, the Vygotskian cultural-historical theory of development postulates that the child’s development begins with the dependence on caregivers (e.g., parents) and occurs in cultural and social contexts, where social interactions are a central part of all human learning [12]. In addition, according to The Tripartite Model of Family Impact on Children’s Adjustment and Emotion Regulation [13], parental characteristics, such as parents’ beliefs about children’s emotions and how parents’ emotion regulation influences children’s social skills.

Therefore, in understanding children’s social skills, we focused on proximal factors such as family, specifically on the parents’ beliefs about children’s emotions and parents’ emotion regulation strategies. Previous research showed that parents’ beliefs about emotions predict mothers’ reactions to children’s negative emotions and children’s emotion regulation [14,15], parents’ expressions of emotions, discussion about emotions in the presence of the children, and children’s coping strategies [16]. In addition, studies that carried out this type of investigation mainly included participants’ mothers or mothers and fathers from different families. Our study includes dyads (mothers and fathers from the same family), aiming to explore the direct and indirect associations of study variables within the whole family system [17]. Specifically, we aimed to investigate at a dyadic level, in a non-clinical preschool children sample, the association between parents’ beliefs about children’s emotions and children’s social skills, and the mediator role of parents’ emotion regulation in this association.

### 1.1. Parents’ Beliefs about Children’s Emotions and Children’s Social Skills

Parents’ beliefs about children’s emotions are widely recognized as important for children’s socialization of emotions, which leads to children’s optimal emotion regulation and social skills [14,18]. In the present study, we focused on two parental beliefs about “children’s anger is valuable” and “children use their emotions to manipulate others” because the two parental beliefs are more salient for parents to endorse among young children. Specifically, previous studies showed that younger children (i.e., preschool children) are more inclined than older children to express anger in front of their parents than with peers, and younger children express more negative emotions compared with older children [19,20]. Therefore, due to the aversive nature of negative emotions, parents are more prone to react, perhaps due to the belief that children use emotions for manipulation [21]. Thus, understanding the manner in which parents endorse parental beliefs about “children’s anger is valuable” and “children use their emotions to manipulate others” is required, considering the implication of children’s anger and manipulation in initiating and maintaining interpersonal relationships [22,23].

Previous studies explored, to a large extent, the association between parents’ beliefs about children’s emotions and parental emotion socialization or children’s socio-emotional outcomes [16,24,25,26]. As far as we know, the direct association between parents’ beliefs about children’s emotions and children’s social skills has been less investigated, with few exceptions [27,28], and even fewer studies have addressed the underlying mechanisms of this association while obtaining data from both parents. Thus, we aim to fill this identified gap in the literature.

Previous studies showed that parents’ beliefs about “children’s anger is valuable”, embedded in parents’ beliefs about ”children’s negative emotions as valuable”, predicted children’s problem-solving skills, support seeking [16], more emotion regulation and less lability [29], and more cooperation and prosocial behavior [27]. Regarding parents’ belief that children use emotions to manipulate others, they are accompanied by parents’ invalidating reactions to children’s positive emotions, which leads to more negative family expressiveness [13,30] and children’s lower social skills. But, previous research showed that positive emotions matter in initiating and regulating social exchanges, and preschoolers who express more positive emotions with peers were more liked one year later [5,31]. Therefore, parents must provide the appropriate level of encouragement for the expression of positive and negative emotions among preschool children, considering that young children depend on their help in regulating emotions and the development of social skills [28,32].

Based on the reviewed studies, we expected an association between parents’ beliefs about children’s emotions and children’s social skills to exist. Therefore, we hypothesized that parents’ belief that “children’s anger is valuable” is positively correlated with children’s social skills (H1), and parents’ belief that “children use their emotions to manipulate others” is negatively correlated with children’s social skills (H2).

### 1.2. Parents’ Emotion Regulation as a Mediator

In general, emotion regulation involves the process through which we influence the emotions we have, and when and how we experience and express them [33]. Regarding emotion regulation in parenthood, it refers to a parent’s ability to control and influence the experience and expression of their emotions in caregiving situations, with cognitive reappraisal and expressive suppression being the most commonly investigated emotion regulation strategies [34,35]. The mediating role of parents’ emotion regulation in the association between parental beliefs about children’s emotion regulation and social skills is warranted in light of two aspects. First, both parents’ beliefs about children’s emotions and parental emotion regulation are influenced by the parents’ beliefs about emotions in general, and by their history and culture [35,36]. Second, parents’ beliefs about emotions are increasingly recognized as an essential aspect of parental socialization of emotions and parenting behaviors more broadly, with implications for children’s development across social, emotional, and behavioral domains [14,16,24,37]. Previous studies showed that parents’ emotion-related beliefs fundamentally influence their emotional socialization behaviors, including their reactions to children’s emotions, talking about emotions, and displaying and expressing their own emotions [38]. The associations between parental beliefs that children’s anger is valuable and parents’ emotion regulation is less studied. However, cognitive reappraisal is associated with reduced negative emotional experience and expression, whereas expressive suppression has been associated with higher levels of negative emotional experience [39]. Therefore, parents who endorse the belief that “children’s anger is valuable” and use cognitive reappraisal may tolerate or alleviate children’s anger, acting as a buffer [40,41]. Consequently, they successfully downregulate anger by reframing, reinterpreting, and modifying any situation in which children’s anger is manifested, an aspect associated with more positive and less negative emotions [29,34]. Therefore, in light of the positive emotional outcomes related to cognitive reappraisal [39], parents who use such emotion regulation strategy are less prone to believe the children use their emotions to manipulate others, considering negative socio-emotional outcomes associated with children’s manipulation behavior (e.g., less prosocial behavior toward peers) [23]. On the other hand, parents who endorse the belief that “children’s anger is valuable” and rely on expressive suppression fail to teach their children how to manage the amount of anger [34]; therefore, according to the modeling hypothesis [42], children, like their parents, tend to suppress their negative emotions, a fact that increases the experience of anger [29]. Among adults, suppression is associated with less positive emotions; therefore, parents using suppression tend to invalidate children’s positive emotions, believing that children use their emotions to manipulate others [30,34].

On the one hand, in Western cultures such as the U.S., or Western Europe, reappraisal is seen as an adaptive emotion regulation strategy, whereas expressive suppression is viewed as a maladaptive response [39,43,44]. For example, Rogers and colleagues found that mothers’ cognitive reappraisal is negatively associated and expressive suppression is positively associated with children’s lability, and expressive suppression is negatively associated with children’s emotion regulation. Further, Xiao and his colleagues [45], using parents’ preschool children as a sample from the New York area, found that parents’ expressive suppression is negatively correlated with children’s positive behavior toward others or their helping behavior. On the other hand, using Chinese preschool children in Hong Kong as a sample, Lau and Williams [46] found that mothers’ use of more reappraisal and less suppression was associated with children’s aggressive behavior. Although previous studies showed existing evidence regarding the indirect association between parental beliefs about children’s emotions and children’s social competence [28], parental emotion regulation was not investigated as a mediator in this association. As emotion regulation strategies, reappraisal and suppression mediate the relation between some beliefs about emotion and emotional outcomes among adults and teenagers, such as successful emotion regulation, well-being, and lower psychological distress [47,48].

Therefore, based on the underlying aspects of both parents’ emotion regulation strategies, we hypothesized that parental cognitive reappraisal mediates the association between parental beliefs about children’s emotions and their social skills. Thus, we anticipated that parents’ belief that “children’s anger is valuable” would positively correlate with parental reappraisal, further predicting children’s social skills (H3), and parents’ belief that “children use their emotions to manipulate others” would negatively correlate with parental reappraisal, further predicting children’s social skills (H4). Further, we hypothesized that parental expressive suppression mediates the association between parental beliefs about preschoolers’ expressed emotions and their social skills. Thus, we anticipated that parents’ belief that “children’s anger is valuable” would negatively correlate with parents’ expressive suppression, further predicting lower levels of children’s social competence (H5) and parents’ belief that “children use their emotions to manipulate others” would positively correlate with parents’ expressive suppression, further predicting lower levels of children’s social competence (H6).

## 2. Materials and Methods

### 2.1. Participants

Our sample comprised 90 dyads of married parents (N = 180 individuals, 90 mothers, and 90 fathers) who participated in the study and completed the questionnaire between 1 September and 30 November 2022. Mothers had a mean age of 34.82 years (SD = 4.7), while fathers had a mean age of 37.46 (SD = 4.7). The children’s ages of each couple ranged from 3 years to 6 years (M = 4.8, SD = 1.2). Regarding the offspring’s gender, this was quite balanced, with 56.7% female and 43.3% male.

The study was approved by the Ethics Committee of the authors’ university. After that, the informed consent was signed and received from all participants. Data were collected using a self-report questionnaire addressed to the parents of the children. Questionnaires were then completed anonymously using Google Forms software (https://www.google.com/forms/about/, accessed on 24 August 2023). Participants were asked to report on the same child (in case they had more than one child), individually complete the questionnaires, and fill in a password (city/town and day/month/year of marriage) to identify the parental dyads. The inclusion criteria for the study were as follows: at least one child between 3 and 6 years old, typically developed.

### 2.2. Measures

Parents’ beliefs about children’s emotions (PBACE) [30] were used to measure the parental beliefs. PBACE has been used in different cultures and assesses multiple beliefs about emotions, being largely invariant across ethnicity and gender [30,49]. The scale has seven dimensions, but in the present study, we used only the value of anger subscale with 6 items, and the manipulation (e.g., “Children often cry just to get attention”) subscale with 4 items. The value of the anger subscale assesses parents’ beliefs that children’s anger is valuable (e.g., “Children’s anger can be a relief to them, like a storm that clears the air”), whereas the manipulation subscale assesses parents’ beliefs that children use expressions of emotion aiming to manipulate others (e.g., “Children often cry just to get attention”). Parents rated statements on 6-points Likert scale, ranging from 1 (strongly disagree) to 6 (strongly agree). The internal consistency of the value of anger subscale was 0.80 for mothers and 0.84 for fathers. For the manipulation subscale, the internal consistency was 0.84 for mothers and 0.85 for fathers.

The Emotion Regulation Questionnaire (ERQ) [34] was used to measure parents’ emotion regulation, specifically to assess the frequency of using cognitive reappraisal and expressive suppression. Cognitive reappraisal describes people who control their emotions by employing cognitive change strategies aiming to modify the emotion–eliciting situation impact, whereas expressive suppression describes people who manage their emotions by inhibiting emotionally expressive behavior [39]. The ERQ has been frequently used in the literature to investigate parents’ outcomes related to children’s socio-emotional development [45,46], and it is reliable for use in research across gender and ethnicity [50]. This self-report scale consists of 10 items measuring an individual’s tendency to use cognitive reappraisal (“When I want to feel less negative emotion, I change what I’m thinking about”) and expressive suppression (“I keep my emotions to myself”) to regulate emotions. Parents rated statements on a 7-point scale, ranging from 1 (strongly disagree) to 7 (strongly agree). In this study, Cronbach’s alpha for the reappraisal subscale was 0.86 for mothers and 0.90 for fathers. Cronbach’s alpha for the suppression subscale was 0.87 for mothers and 0.82 for fathers.

Preschool and Kindergarten Behavior Scale-2 (PKBS-2) [51] was used to measure children’s social skills, which is used in different cultures. The34-item scale *assesses* the desirable social skills that are indicators of well-adapted children between 3 and 6 years old, which are grouped into three dimensions: social cooperation (e.g., “Is cooperative”), social interactions (e.g., “Ask for help from adults when is needed”) and social independence (e.g., “Is invited by other children to play”). Parents rated statements using a 4-point scale from never (0) to often (4). In this study, Cronbach’s alpha for the social skills subscale was 0.95 for mothers and 0.96 for fathers. Additionally, we assessed the following demographic variables: parents’ and children’s age and gender.

Children’s social skills, the outcome of interest of our study, did not vary according to children’s gender; this variable was considered a covariate in previous studies [27]. Therefore, children’s gender was not included as a covariate in further analyses. Similarly, children’s social skills did not vary according to their parents’ and children’s age. Thus, these variables were not included as covariates in the subsequent analyses.

### 2.3. Statistical Analysis

We used both SPSS 26.0 and Amos 26.0 programs [52] for the descriptive statistics and the estimation of the direct and indirect effects of mediation at the dyadic level. We used the common fate mediation model (CFM) [53] due to the similarities among parents regarding their children [54]. To test the CFM models, we used structural equation modeling (SEM). We verified different fit indices, such as the chi-square test, whose value is acceptable when it is not significant; the goodness of fit index (GFI); the adjusted goodness of fit index (AGFI); the comparative fit index (CFI); the normed fit index (NFI); and the root mean square error of approximation (RMSEA). The significance of the indirect effects and standard errors were tested using Sobel’s formula [55].

## 3. Results

### 3.1. Preliminary Analyses

Table 1 shows no significant differences between mothers’ and fathers’ reports on cognitive reappraisal. However, mothers reported stronger beliefs that children’s anger is valuable compared to fathers. On the other hand, fathers reported stronger beliefs that children’s emotions are manipulative compared to mothers, and fathers reported higher expressive suppression and lower levels of children’s social skills compared to mothers. The effect sizes found were small (0.22, −0.26, 0.12, −0.33, and 0.34).

Correlations between the study’s main variables and the intercorrelations between mothers’ and fathers’ reports were computed (Table 2).

The results in Table 2 show negative correlations between parents’ beliefs that children’s anger is valuable and children’s social skills (mothers: *r* = −0.46, *p* < 0.001; fathers: *r* = −0.49, *p* < 0.001) and between parents’ beliefs that children’s emotions are manipulative and children’s social skills (mothers: *r* = −0.46, *p* < 0.001; fathers: *r* =−0.37, *p* < 0.001). Negative correlations were also found between parents’ beliefs that children’s anger is valuable and parents’ cognitive reappraisal (mothers: *r* = −0.29, *p* < 0.01; fathers: *r* = −0.44, *p* < 0.001), and negative correlations were also found between parents’ beliefs that children’s emotions are manipulative and parents’ cognitive reappraisal (mothers: *r* = −0.21, *p* < 0.05; fathers: *r* = −0.31, *p* < 0.01). Positive correlations were found between parents’ cognitive reappraisal and children’s social skills for both parents (mothers: *r* = 0.53, *p* < 0.001; fathers: *r* = 0.41, *p* < 0.001). A negative correlation was found between parents’ cognitive suppression and children’s social skills, but only for mothers (*r* = −0.26, *p* < 0.05).

### 3.2. Testing the CFMs

#### 3.2.1. The CFM of the Value of Anger Parental Belief and Children’s Social Skills, with Parents’ Cognitive Reappraisal as Mediator

We examined a CFM model in which the value of parental belief about anger predicts children’s social skills through the mediation of parents’ cognitive reappraisal (Figure 1). This model has a good fit (*χ*^2^(6) = 4.12; *p* = 0.65; GFI = 0.98; AGFI = 0.94; NFI = 0.98; CFI =1.00; RMSEA = 0.00). The results show that stronger parental beliefs about the value of anger are associated with lower levels of parents’ cognitive reappraisal (*β* = −0.65, *p* < 0.001). Parents’ cognitive reappraisal is positively associated with children’s social skills (*β* = 0.35, *p* < 0.05). Moreover, after introducing the mediator, the value of parental anger belief remains negatively associated with lower levels of children’s social skills (*β* = −0.55, *p* < 0.01). The model explains 70% of the variance in children’s social skills, with a total standardized effect of −0.78 (of parents’ beliefs that children’s anger is valuable for children’s social skills), split into a direct standardized effect of −0.55 and an indirect standardized effect through parents’ cognitive reappraisal of −0.23). To conclude, parents’ cognitive reappraisal was found to partially mediate the relationship between parents’ beliefs that children’s anger is valuable and children’s social skills (*β* = −0.23, *p* < 0.05).

#### 3.2.2. The CFM of the Parental Beliefs Regarding Children’s Emotions as Manipulative and Children’s Social Skills, with Parents’ Cognitive Reappraisal as Mediator

We examined a CFM model in which parental beliefs about children’s emotions as manipulative predict children’s social skills through the mediation of parents’ cognitive reappraisal (Figure 2). This model provides a good fit (*χ*^2^(6) = 3.98; *p* = 0.67; GFI = 0.98; AGFI = 0.95; NFI = 0.98; CFI = 1.00; RMSEA = 0.00). The results show that stronger parental beliefs that children’s emotions are manipulative are associated with lower cognitive reappraisal of parents (*β* = −0.43, *p* < 0.01), and parents’ cognitive reappraisal is positively associated with children’s social skills (*β* = 0.51, *p* < 0.001). Moreover, after introducing the mediator, higher levels of parents’ beliefs that children’s positive emotions are manipulative were still associated with lower levels of children’s social skills (*β* = −0.46, *p* < 0.001). The model explains 70% of the variance in children’s social skills, with a total standardized effect of −0.67 (of parents’ beliefs that children’s emotions are manipulative on children’s social skills), split into a direct standardized effect of −0.46 and an indirect standardized effect through parents’ cognitive reappraisal of −0.21. To conclude, parents’ cognitive reappraisal was found to partially mediate the relationship between parents’ beliefs that children’s emotions are manipulative and children’s social skills (*β* = −0.21, *p* < 0.01).

Regarding the CFM with parents’ expressive suppression as a mediator of the association between the value of anger and preschoolers’ social skills, although the model provides a good fit (*χ*^2^(6) = 7.02; *p* = 0.31; GFI = 0.97; AGFI = 0.91; NFI = 0.95; CFI = 0.99; RMSEA = 0.04), parents’ expressive suppression is not significantly associated with children’s social skills (*β* = −0.02, *p* = 0.98). Similarly, regarding the CFM with parents’ expressive suppression as a mediator of the association between parents’ beliefs that preschoolers’ emotions are manipulative and children’s social skills, although the model provides an acceptable good fit (*χ*^2^(6) = 11.12; *p* = 0.08; GFI = 0.96; AGFI = 0.86; NFI = 0.92; CFI = 0.96; RMSEA = 0.09), parents’ expressive suppression is not significantly associated as a mediator with children’s social skills (*β* = 0.06, *p* = 0.67).

## 4. Discussion

In this study, we examined the association between parents’ beliefs about children’s emotions and children’s social skills at a dyadic level in a non-clinical preschool children sample. Also, we aimed to investigate whether parents’ emotion regulation strategies play a mediating role in the relationship between parents’ beliefs about children’s emotions and children’s social skills via CFM.

Contrary to expectations, our findings indicate a direct and negative association between parents’ belief that “children’s anger is valuable” and children’s social skills. Therefore, the first hypothesis was not supported. But, the obtained results are explainable to some extent. Based on the fact that the parental belief “children’s anger is valuable” is linked with two types of emotional socialization, namely negative family expressiveness and parents’ reactions to children’s emotions, previous research reported mixed findings regarding their association with children’s social skills. Although previous studies showed a positive association between parents’ beliefs about ”children’s negative emotions as valuable” and children’s socio-emotional outcomes [16,29], parents who value children’s anger were more negatively expressive and more supportive of their children’s negative emotions [30]. Therefore, children exposed to high levels of parental expression of negative emotions may experience negative emotional overarousal, which makes it more difficult to maintain social interaction exchanges or solve conflicts with peers [56,57]. Further, based on the social learning theory, according to which children learn through modeling, reinforcement, and imitation [8,10], the predominant negative family expressiveness stems from the parental beliefs that “children’s anger is valuable is detrimental to the children’s social skills. Specifically, parents who express negative emotions, such as anger, could encourage their children to display the same negative emotional expression [9], and previous studies showed that negative emotions, especially anger, are less tolerated in social interactions [14]. For example, negative emotions, including anger, among kindergarten girls predicted lower peer acceptance later [58], and in a study using a short-term longitudinal design, kindergartners’ anger frequency also predicted lessened peer acceptance and disputes with teachers [59]. Further, regarding parents’ reactions to their children’s emotions, previous studies showed that parents who value children’s negative emotions, such as anger, manifest supportive reactions by encouraging children to express and discuss negative emotions [16,28]. Therefore, this result contradicts those obtained by Wong and colleagues [28], who showed that moderately high levels of encouragement of children’s negative emotional expression and moderate levels of family negative expressiveness were related to higher children’s social competence as reported by peers. Another explanation could be offered by the fact that parents may be more aware that children’s anger could turn into aggressive behavior [60], trying to prevent such behaviors in this manner. Moreover, one possible explanation for this finding may be that in the previous studies that investigated the implications of parents’ beliefs about “children’s anger is valuable” on parents’ perception of children’s social skills, the authors, while measuring parents’ beliefs about “children’s negative emotions as valuable”, included other negative emotion besides anger, such as sadness, which is easier to manage compared to anger. Hence, for children, managing anger is more difficult to handle, which is perceived as more salient, threatening and involving children’s activation of self-protective mechanism [9,61]. As such, our unexpected results underline the recognition of anger as a negative, salient and dominant emotion among young children [9].

Our findings support the second hypothesis that parents’ belief that “children use their emotions to manipulate others” is negatively correlated with children’s social skills. Halberstadt and colleagues [30] noted that parents who believe that children use emotions to manipulate others reported less validating and more invalidating reactions to children’s positive emotions. Indeed, previous research showed that positive emotions matter in initiating and regulating social exchanges, invalidating reactions to children’s positive emotions with problematic consequences on social interactions [5].

Overall, our results regarding the first two hypotheses contradict previous research [27,62] in which parental variables had no direct effects on children’s socio-emotional outcomes.

Further, we hypothesized that parents’ belief that “children’s anger is valuable” would positively correlate with parental reappraisal, further predicting children’s social skills. Contrary to our hypothesis, the results indicate that parents’ belief that “children’s anger is valuable” is associated with lower levels of parents’ reappraisal, which is further positively associated with children’s social skills. The negative association between parents’ belief that “children’s anger is valuable” and parents’ cognitive reappraisal could be explained by previous studies showing that negative dominant expressivity, which arises from parents’ belief that “children’s anger is valuable”, is negatively associated with reappraisal [14,30], which in turn is associated with higher children’s emotion regulation competence, a precursor of children’s social skills [13,63]. Moreover, parents’ cognitive reappraisal is associated with experiencing and expressing positive emotions to a greater extent, and by diminishing negative emotions, parents aim to decrease undesired emotions among their children [64,65].

Further, we anticipated that parents’ belief that “children use their emotions to manipulate others” would negatively correlate with parental reappraisal, further predicting children’s social skills. The fourth hypothesis was supported. Responding to and reacting to children’s emotions can be exhausting, stressful, and overwhelming for children’s parents [66]. Therefore, parents who cannot manage the amount of children’s emotional expression and experiences, and those who are not aware of children’s emotional needs and development, are less likely to reappraise children’s unpleasant or undesirable emotional responses, further endorsing the belief that ”children use their emotions to manipulate others” [30,40,63]. Moreover, the negative association between parents’ beliefs about emotions as manipulative and parents’ cognitive reappraisal is explained by the fact that parents’ use of reappraisal endorses more supportive reactions toward children since they believe children’s emotions are uncontrollable. Therefore, young children do not have control over their emotions and, consequently, do not aim to use them for manipulative purposes [30,66]. Further, correlational studies also indicate that higher levels of parents’ cognitive reappraisal are associated with higher levels of children’s social competence, including social skills, such as prosocial behavior [46]. Specifically, previous studies showed that individuals who typically use reappraisal report having closer relationships with friends and more social skills [34]. Consistent with a modeling hypothesis [13], mothers with cognitive reappraisal strategies have children who utilize the same emotion regulation strategy [67]. Consequently, we can presume that children’s cognitive reappraisal promotes children’s social skills development.

Further, we hypothesized that parental expressive suppression mediates the association between parental beliefs about preschoolers’ expressing emotions and their social skills. Thus, we anticipated that parents’ belief that “children’s anger is valuable” would negatively correlate with parents’ expressive suppression, which will further predict lower levels of children’s social competence, and parents’ belief that “children use their emotions to manipulate others” would positively correlate with parents’ expressive suppression, which will further predict lower levels of children’s social competence. The fourth and the sixth hypotheses were not supported. A possible explanation is that suppressors have less emotionally close relationships [68] and weaker interpersonal relationships [69,70]. Moreover, using suppression interferes with consolidating close social connections [71]. As such, given that the parent–child relationship is the earliest and closest during early childhood, and the most enduring of all interpersonal bonds [72], fewer parents use expressive suppression strategy excessively, except those from risk families [73]. Finally, the lack of the mediational role of parents’ expressive suppression in the associations mentioned above may be caused by the children’s age; parents of younger children are likely to perceive them as more emotionally vulnerable than older children [21]. Therefore, parents may employ less expressive suppression when responding to children’s emotions.

Some limitations should be also discussed. First, due to our study’s cross-sectional nature, we cannot establish causal associations among the variables. Thus, taking into account that children’s emotional needs change across different developmental, the same parental variables (i.e., parental beliefs about children’s emotions), endorsed at different ages of children, may have various consequences on children’s socio-emotional development [74]; thus, studies with longitudinal design are required. Also, in line with the previous comment, future studies could explore the moderator role of children’s age in the association between parents and children’s variables. Second, we did not take into consideration other related variables regarding children (e.g., temperament and emotionality), related variables regarding parents (e.g., parenting practices), or kindergarten variables (e.g., teachers’ reports, interviews with teachers, and direct observation of the class). Therefore, we were not able to directly observe children and assess the variables, which may account for the association between parents’ beliefs about children’s emotions and children’s social skills, and which may capture children’s social skills more accurately. Therefore, future studies could benefit from investigating such variables by integrating direct assessment measures Third, because cultural perspective accounts for understanding different children’s socio-emotional outcomes, future studies might benefit from exploring parents’ and children’s variables in other cultural backgrounds. Fourth, although we integrated mothers’ and fathers’ perspectives, our sample size is relatively small. Thus, future studies should investigate the relationship between variables using more dyads.

Also, our findings pointed out some clinical implications, targeting children’s parents and kindergartens. Parents should be more aware of the children’s emotional needs, and, according to the Vygotskian approach (e.g., socio-cultural framework and interconnectedness), in promoting children’s social skills, they have to initiate activities aiming to target the inter-psychological dimension at first (“child-adult” interaction), followed by intra-psychological dimension (the development within the child) whereby children may experience and express emotions [75]. Also, kindergarten might create multiple education contexts (engage children in social interaction with peers through play) to promote children’s social skills development. In addition, preschool psychologists should provide support to the parents and children, aiming to promote the understanding of parental beliefs regarding children’s emotions and to emphasize positive outcomes related to the parents’ use of cognitive reappraisal on children’s social skills development.

## 5. Conclusions

The present study highlights the dynamic character of parental influences on children’s social outcomes, which is explained through the lens of the tripartite model of family impact on children’s adjustment and emotion regulation, using both mothers’ and fathers’ perceptions. We highlighted that specific parents’ beliefs about children’s emotions, directly and indirectly, influence children’s social competence through the mediating role of parents’ emotion regulation. The findings emphasize the need to implement preventive intervention in promoting children’s social skills development in kindergartens, considering the positive outcomes associated with children’s social competence. Furthermore, restructuring parents’ beliefs regarding children’s emotions and promoting parents’ cognitive reappraisal as an adaptive emotion regulation strategy is required.

## Figures and Tables

**Figure 1 children-10-01473-f001:**
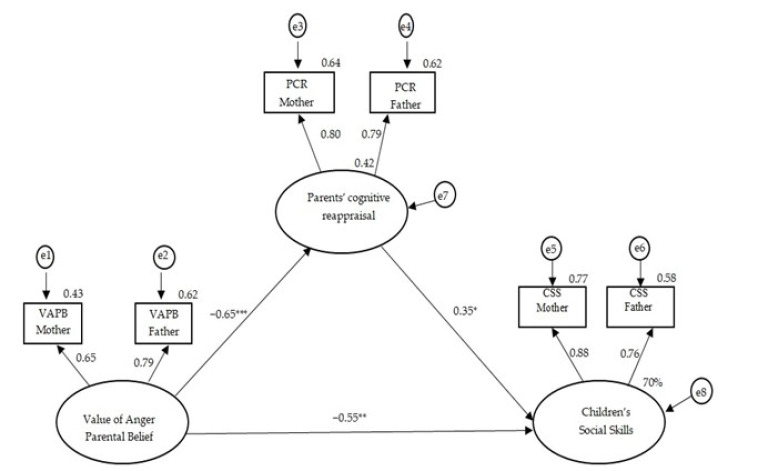
The common fate model testing the association between parents’ beliefs regarding the value of anger and children’s social skills, mediated by parents’ cognitive reappraisal. * *p* < 0.05. ** *p* < 0.01. *** *p* < 0.001.

**Figure 2 children-10-01473-f002:**
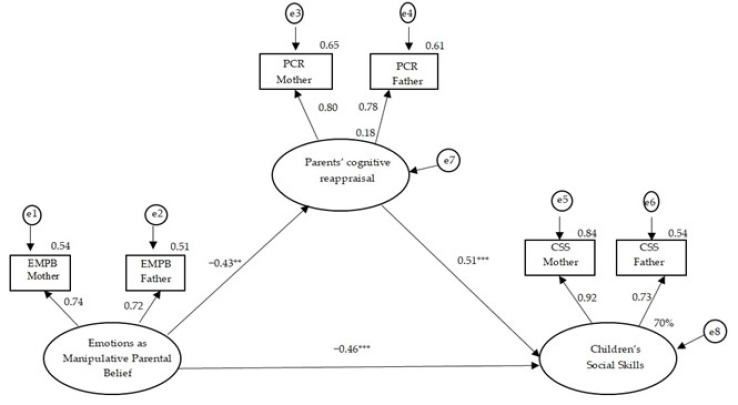
The common fate model testing the association between parents’ belief about emotions as manipulative and children’s social skills, mediated by parents’ cognitive reappraisal. ** *p* < 0.01. *** *p* < 0.001.

**Table 1 children-10-01473-t001:** Means, standard deviations, and paired-samples *t*-test.

	Mothers		Fathers			
Variable	Mean	SD	Mean	SD	T	d
Children’s anger is valuable	23.04	6.27	21.62	6.72	2.11 *	0.22
Children’s emotions are manipulative	15.51	5.33	16.84	5.06	−2.51 *	−0.26
Parents’ cognitive reappraisal	4.89	1.20	4.76	1.28	1.17	0.12
Parents’ expressive suppression	3.40	1.39	3.91	1.30	−3.12 **	−0.33
Children’s social skills	79.61	15.95	74.76	18.24	3.61 **	0.34

Note: * *p* < 0.05; ** *p* < 0.01.

**Table 2 children-10-01473-t002:** Correlations between study variables and between mothers’ and fathers’ responses.

Variable	1	2	3	4	5
1. Children’s anger is valuable	**0.51 *****	0.35 ***	−0.29 **	0.04	−0.46 ***
2. Children’s emotions are manipulative	0.58 ***	**0.53 *****	−0.21 *	0.44 **	−0.46 ***
3. Parents’ cognitive reappraisal	−0.44 ***	−0.31 **	**0.63 *****	0.04	0.53 ***
4. Parents’ expressive suppression	−0.00	0.06	0.37 ***	**0.33 ****	−0.26 *
5. Children’s social skills	−0.49 ***	−0.37 ***	0.41 ***	−0.09	**0.67 *****

Note: Correlations are shown above the diagonal for the mothers and below the diagonal for the fathers. Intercorrelations between the parents are shown along the diagonal in bold. Note: * *p* < 0.05; ** *p* < 0.01; *** *p* < 0.001.

## Data Availability

The data that support the findings of this study are available upon request from the corresponding author due to privacy issues.

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
