# Peer review of "Parents’ Beliefs about Children’s Emotions and Children’s Social Skills: The Mediating Role of Parents’ Emotion Regulation"

_children, 2023, doi:10.3390/children10091473_

Round 1
Reviewer 1 Report
Dear authors! I have read your paper with great interest. Your study is very relevant. I suggest to check cultural historical theory of human development by L.S. Vigotsky and his followers for your background support. The thing is that there are no just emotions without the gola of activity and child's participation in joint activity with an adult. So, the point is that it is not just expression of a kind of emotion, but the specific situation or activity, in which this emotion is expressed. Another aspect is the absence of direct assessment of the children, you take into account only the parents opinions and beliefs. It is very important, but if you are talking about some kind of effects, it is not so precise. You may say about these beliefs and opinion for sure, but you may not say that it is "an affect". Som it is not possible to talk about real skills of the children and only about the parent's beliefs. In order to know something clear about the children, it would be necessary to interview the teachers and also, very important, to organize psychological assessment of the children or direct observation of the class. Please, take this aspects into account in your discussion and in conclusions.
Author Response
Point 1: Thank you for the important suggestions! We have considered each recommended suggestion.
Dear authors! I have read your paper with great interest. Your study is very relevant. I suggest to check cultural historical theory of human development by L.S. Vigotsky and his followers for your background support. The thing is that there are no just emotions without the goal of activity and child's participation in joint activity with an adult. So, the point is that it is not just expression of a kind of emotion, but the specific situation or activity, in which this emotion is expressed. Another aspect is the absence of direct assessment of the children, you take into account only the parents opinions and beliefs. It is very important, but if you are talking about some kind of effects, it is not so precise. You may say about these beliefs and opinion for sure, but you may not say that it is "an affect". Some it is not possible to talk about real skills of the children and only about the parent's beliefs. In order to know something clear about the children, it would be necessary to interview the teachers and also, very important, to organize psychological assessment of the children or direct observation of the class. Please, take this aspects into account in your discussion and in conclusions.
Response 1: We want to thank Reviewer #1 for these suggestions. Regarding the first opinion about the relevance of the cultural-historical theory of human development by L.S. Vygotsky, we added the following paragraphs:”Further, the Vygotskian cultural-historical theory of development postulates that the child’s development begins with the dependence on caregivers (e.g., parents) and occurs in cultural and social contexts, where social interactions are a central part of all human learning (John-Steiner & Mahn, 1996).” (lines 44-47) and “…according to the Vygotskian approach (e.g., socio-cultural framework, interconnectedness), in promoting children’s social skills, they have to initiate activities aiming to target first the inter-psychological dimension (“child-adult” interaction), followed by intra-psychological dimension (the development within the child) whereby children may experience and express emotions (Rubtsov, 2020).” (lines 457-461). Regarding the second opinion, we highlight the importance of psychological assessment of the children or direct observation of the class, seen as a limitation of our study (lines 444-448), and we propose a future research direction that will capture effective children’s social skills through direct assessment measures (lines 448-450).

Reviewer 2 Report
Just a few remarks:
l. 185: Provide an overview of all of the items.
l. 204: 2x 'by other'
l. 333: contradicts
l. 406: I would have expected analyses with parents'and child's age. Why not?
Author Response
Point 1. 185: Provide an overview of all of the items.
Response 1: We added more information concerning all the items of measures:
Regarding the Parents' beliefs about children's emotions (PBACE), we added: „The PBACE has been used in different cultures and assesses multiple beliefs about emotions, being largely invariant across ethnicity and gender (Caiado et al., 2021; Halberstadt et al., 2013).” (lines 191-192)” and “The Value of anger subscale assesses parents’ beliefs that children’s anger is valuable (e.g., „ Children’s anger can be a relief to them, like a storm that clears the air”), whereas the Manipulation subscale assesses parents’ beliefs that children use expressions of emotion aiming to manipulate others (e.g., ,, Children often cry just to get attention”)” (lines 196-200).
Regarding the Emotion Regulation Questionnaire (ERQ), we added: “…specifically to assess the frequency of using cognitive reappraisal and expressive suppression. Cognitive reappraisal describes people who control their emotions by employing cognitive change strategies aiming to modify the emotion–eliciting situation impact, whereas expressive suppression describes people who manage their emotions by inhibiting emotion–expressive behaviour (John & Gross, 2004). The ERQ has been frequently used in the literature, investigating the parents’ outcomes related to children’s socio-emotional development (Lau & Williams, 2022; Xiao et al., 2018), being reliable to use in research across gender and ethnicity (Melka et al., 2011). (lines 205-212)
Regarding the Preschool and Kindergarten Behavior Scale-2 (PKBS-2), we added: “...being used in different cultures (Carapito et al., 2018). The 34–item scale assesses the desirable social skills that are indicators of well-adapted children between 3 and 6 years old, the items being grouped into three dimensions: social cooperation (e.g., “ Is cooperative”), social interactions (e.g., “Ask for help from adults when is needed”), and social independence (e.g., “Is invited by other by other 204 children to play”). (lines 220-224)
Point 2.204: 2x 'by other'
Response 2: We removed one “by other”. (line 224)
Point 3.333: contradicts
Response 3: We change "contradict" with " contradicts ". (line 358)
Point 4.406: I would have expected analyses with parents' and child's age. Why not?
Response 4:Thank you for your observation. The preliminary correlational analyses showed that children’s social skills, the outcome of interest of our study, did not vary by parents’ and children’s age. Moreover, regarding children’s age, our result aligns with Xiao and colleagues’ (2018) findings, according to which the prosocial behavior, an indicator of children’s social skills, was unrelated to children’s age. As such, our analyses did not include the parent's and child's age as covariates. We added this information regarding analyses with parents' and children's age:“Similarly, children’s social skills did not vary by parents’ and children’s age. Thus, these variables were not included as covariates in subsequent analyses.” (lines 233-235 )
In addition, regarding analyses including child's age we mentioned that the child’s age could be investigated in future studies as a moderator variable in the association between parents and children's variables. (lines 440 - 442)
